# The GTPase Rab26 links synaptic vesicles to the autophagy pathway

Beyenech Binotti[1†], Nathan J Pavlos[2†], Dietmar Riedel[3], Dirk Wenzel[3], Gerd Vorbrüggen[4,5], Amanda M Schalk[1‡], Karin Kühnel[1], Janina Boyken[1§], Christian Erck[6], Henrik Martens[6], John JE Chua[1*], Reinhard Jahn[1*]

[1]Department of Neurobiology, Max Planck Institute for Biophysical Chemistry, Göttingen, Germany; [2]School of Surgery, University of Western Australia, Crawley, Australia; [3]Facility for Transmission Electron Microscopy, Max Planck Institute for Biophysical Chemistry, Göttingen, Germany; [4]Research Group Molecular Cell Dynamics, Max Planck Institute for Biophysical Chemistry, Göttingen, Germany; [5]Department of Developmental Biology, Georg-August-University Göttingen, Göttingen, Germany; [6]Synaptic Systems GmbH, Göttingen, Germany

*For correspondence: john.
chua@mpibpc.mpg.de (JJC);
rjahn@gwdg.de (RJ)

†These authors contributed
equally to this work

Present address: ‡Department of
Biochemistry and Molecular
Genetics, University of Illinois at
Chicago, Ashland, United States;
§Gynecological Therapies, Bayer
Pharma Aktiengesellschaft,
Berlin, Germany

Competing interests: See
page 18

Reviewing editor: Christian
Rosenmund, Charité-
Universitätsmedizin Berlin,
Germany

**Abstract** Small GTPases of the Rab family not only regulate target recognition in membrane traffic but also control other cellular functions such as cytoskeletal transport and autophagy. Here we show that Rab26 is specifically associated with clusters of synaptic vesicles in neurites. Overexpression of active but not of GDP-preferring Rab26 enhances vesicle clustering, which is particularly conspicuous for the EGFP-tagged variant, resulting in a massive accumulation of synaptic vesicles in neuronal somata without altering the distribution of other organelles. Both endogenous and induced clusters co-localize with autophagy-related proteins such as Atg16L1, LC3B and Rab33B but not with other organelles. Furthermore, Atg16L1 appears to be a direct effector of Rab26 and binds Rab26 in its GTP-bound form, albeit only with low affinity. We propose that Rab26 selectively directs synaptic and secretory vesicles into preautophagosomal structures, suggesting the presence of a novel pathway for degradation of synaptic vesicles.

## Introduction

Synapses are highly dynamic structures exhibiting frequent turnover. The most dramatic phase of synaptic remodeling occurs during development when the majority of initially formed synapses are eliminated while the final synaptic network is being generated. However, even in the adult brain there is persistent turnover of synapses, mostly in response to experience and learning (*Caroni et al., 2012*; *Chung and Barres, 2012*).

Formation of a new synapse involves the establishment of highly specialized structures containing arrays of unique membrane and scaffold proteins, which necessitates close coordination between the presynaptic axon and the postsynaptic dendrite. Components of these structures are delivered by microtubule-based transport, although some of the proteins are locally synthesized in dendrites (*Steward and Levy, 1982*; *Holt and Schuman, 2013*). Similarly, delivery of synaptic membranes (such as synaptic vesicles) and active zone precursors to the synapse relies on kinesin-mediated transport (*Hirokawa et al., 2010*). A lot has been learned in recent years about the signaling events and the downstream effectors involved in synaptogenesis as well as the mechanisms by which individual components are recruited and maintained (*Caroni et al., 2012*).

Less is known, however, about the molecular cascades involved in synaptic elimination. In principle, two mechanisms of elimination co-exist that are not mutually exclusive and may indeed be coordinated with each other: 'attack from the outside' or 'self-destruction from the inside'. Elimination from the

**eLife digest** Our brain contains billions of cells called neurons that form an extensive network through which information is readily exchanged. These cells connect to each other via junctions called synapses. Our developing brain starts off with far more synapses than it needs, but the excess synapses are destroyed as the brain matures. Even in adults, synapses are continuously made and destroyed in response to experiences and learning.

Inside neurons there are tiny bubble-like compartments called vesicles that supply the synapses with many of the proteins and other components that they need. There is a growing body of evidence that suggests these vesicles are rapidly destroyed once a synapse is earmarked for destruction, but it is not clear how this may occur.

Here, Binotti, Pavlos et al. found that a protein called Rab26 sits on the surface of the vesicles near synapses. This protein promotes the formation of clusters of vesicles, and a membrane sometimes surrounds these clusters. Further experiments indicate that several proteins involved in a process called autophagy—where unwanted proteins and debris are destroyed—may also be found around the clusters of vesicles.

Autophagy starts with the formation of a membrane around the material that needs to be destroyed. This seals the material off from rest of the cell so that enzymes can safely break it down. Binotti, Pavlos et al. found that one of the autophagy proteins—called Atg16L—can bind directly to Rab26, but only when Rab26 is in an 'active' state.

These findings suggest that excess vesicles at synapses may be destroyed by autophagy. Further work will be required to establish how this process is controlled and how it is involved in the loss of synapses.

outside is usually executed by microglial cells but the underlying signaling network is complex, and astrocytes have also recently been appreciated to play a major role in this process (*Chung and Barres, 2012*; *Stephan et al., 2012*; *Clarke and Barres, 2013*; *Maor-Nof and Yaron, 2013*). In cell autonomous elimination, synaptic components may either be (i) recycled, that is, being removed in a functionally intact form for the use at another site, or (ii) degraded. With the exception of some recent evidence showing that synaptic vesicles can be exchanged between neighboring synapses, whether synaptic components can be reused after having been operational in a functional synapse remains largely unexplored (*Darcy et al., 2006*; *Kamin et al., 2010*). Conversely, an increasing body of evidence supports the view that synaptic components are rapidly degraded once a synapse is earmarked for elimination. Unsurprisingly, the ubiquitin-proteasome system is emerging as one of the central players, both at presynaptic (*Yao et al., 2007*; *Jiang et al., 2010*) and postsynaptic sites ([*Bedford et al., 2001*; *Patrick et al., 2003*; *Lee et al., 2004*], for a review see *Mabb and Ehlers (2010)*). At the presynaptic site, the ubiquitin system is not only involved in synaptic elimination, but also in the general regulation of synaptic plasticity (*Muralidhar and Thomas, 1993*; *Campbell and Holt, 2001*; *DiAntonio et al., 2001*; *Murphey et al., 2003*; *Yao et al., 2007*; *Yi and Ehlers, 2007*; *Lee et al., 2008*). For instance the protein RIM, a crucial hub for organizing active zones that form the release site for synaptic vesicles, was recently shown to be rapidly turned over upon ubiquitination, resulting in loss of synaptic function (*Yao et al., 2007*). Furthermore, an increasing number of ubiquitin-modifying enzymes have been described from synapses (particularly E3-ligases) (*Ding and Shen, 2008*).

In contrast to the emerging role of the ubiquitin proteasome system, little information is currently available regarding the mechanisms by which synaptic membrane proteins are eliminated. At the postsynaptic site, ubiquitin-dependent pathways are clearly involved in the regulation of surface receptor density (*Patrick et al., 2003*; *Kato et al., 2005*; *Schwarz et al., 2010*). However, only scant information is available about the turnover of membrane proteins at the presynaptic site where a complicated and autonomous vesicle recycling machinery needs to deal with many 100s of synaptic vesicles. Surprisingly, the mechanisms by which synaptic vesicles are eliminated have thus far received little attention. By analogy to non-neuronal cells, it is frequently assumed that synaptic vesicle membrane proteins follow the canonical endosomal-lysosomal route for degradation which involves ubiquitination and recognition by the ESCRT machinery after being delivered to endosomes, followed by the formation of multivesicular bodies, retrograde transport, and ultimately fusion with lysosomes

(*Katzmann et al., 2002*; *Raiborg and Stenmark, 2009*). However, aside from a few hints from a recent proteomic study, whether synaptic vesicle proteins are ubiquitinated remains unclear (*Na et al., 2012*). Similarly, whether sequestration into the lumen of multivesicular bodies is involved and, if so, to what extent is unknown. Indeed, multivesicular bodies are infrequently observed in axons and typically appear in response to pathological (dystrophic or toxic) conditions (for review see [*Von Bartheld and Altick, 2011*]). Furthermore, no information is currently available concerning the involvement of the ESCRT pathway in the elimination of presynaptic components. Intriguingly, recent studies implicate the involvement of clathrin-dependent pathways in targeting plasma membrane components to autophagosomes, hinting at the potential involvement of this mechanism in the turnover of synaptic vesicles recovered by endocytosis following the release of their neurotransmitter content (*Ravikumar et al., 2010*).

In this study we report about data suggesting the presence of a novel pathway for the degradation of synaptic and secretory vesicles, which involves selective sequestration of vesicle clusters into structures resembling early autophagosomes. This pathway is triggered by Rab26, a little-studied member of the Rab-GTPase superfamily that is related to the exocytotic Rab3/ Rab27 subgroup. We show that Rab26 selectively localizes to presynaptic membrane vesicles and recruits both Atg16L1 and Rab33B, two components of the pre-autophagosomal machinery. Remarkably, these autophagosomal structures are filled almost exclusively with synaptic vesicles and proteins typically associated with large dense-core vesicles. Overexpression of EGFP-tagged Rab26, but not of FLAG-tagged or wild-type (WT) Rab26, induces the formation of giant autophagosomes in the cell bodies of hippocampal neurons—a phenotype that is mirrored upon transfection in HeLa cells. Based on these findings, we conclude that Rab26 may selectively channel synaptic vesicles into pre-autophagosomes and, thus, may represent a new regulator of synapse turnover.

## Results

### Rab26 is a GTPase enriched on synaptic vesicles with properties reminiscent of Rab27b

Previously we reported that synaptic vesicles highly purified from rat brain contain more than 30 different Rab-GTPases (*Takamori et al., 2006*). Of these, a subgroup of Rabs including Rab3a, Rab3b, Rab3c, and Rab27b were highly enriched in the vesicle fraction (*Pavlos et al., 2010*). These Rabs are known to function in the regulation of exocytosis and constitute part of the 'secretory Rab subfamily' (*Pereira-Leal and Seabra, 2001*; *Fukuda, 2008*; *Pavlos and Jahn, 2011*). Rab26, a comparatively uncharacterized member of the Rab superfamily, is also closely related to this subgroup. Since we detected Rab26 on purified synaptic vesicles in two previous independent proteomic studies (*Takamori et al., 2006*; *Pavlos et al., 2010*), we decided to further explore its endogenous localization in neurons. To this end, we raised a mouse monoclonal antibody that is specific for Rab26 and does not cross-react with other related Rab proteins including Rab27 (*Figure 1—figure supplement 1*).

First, we used immunoblotting to monitor the subcellular distribution of Rab26 during the purification of synaptic vesicles from the rat brain. As shown in *Figure 1A*, Rab26 co-purified with synaptic vesicle markers (as indicated here using synaptophysin), with the highest enrichment being observed in the synaptic vesicle (SV) fraction obtained after purification using consecutive density gradients and size exclusion chromatography. This fraction has been previously shown to be comprised almost exclusively of synaptic vesicles (at least 95% purity) (*Huttner et al., 1983*). For independent confirmation, we carried out immunoisolation of synaptic vesicles using beads (Eupergit C1Z) covalently coupled with monoclonal antibodies specific for Rab26 or synaptophysin. As shown in *Figure 1B*, both antibodies resulted in the isolation of membranes highly enriched in both synaptophysin and Rab26. As a control, the membranes were solubilized with the detergent Triton X-100 prior to immunoisolation (Tx-IP). In this case, only the respective antigens were isolated (*Figure 1B*), thus validating the specificity of the isolation procedure. We also verified the nature of the immunoisolated vesicles by transmission electron microscopy (TEM). As previously reported, synaptophysin-beads were densely covered by small vesicular profiles with a size distribution typical for synaptic vesicles (i.e., 40–45 nm) (*Figure 1C*) (*Burger et al., 1989*; *Takamori et al., 2000*). Rab26 beads were similarly populated with these vesicles, albeit to a lesser extent (*Figure 1D*).

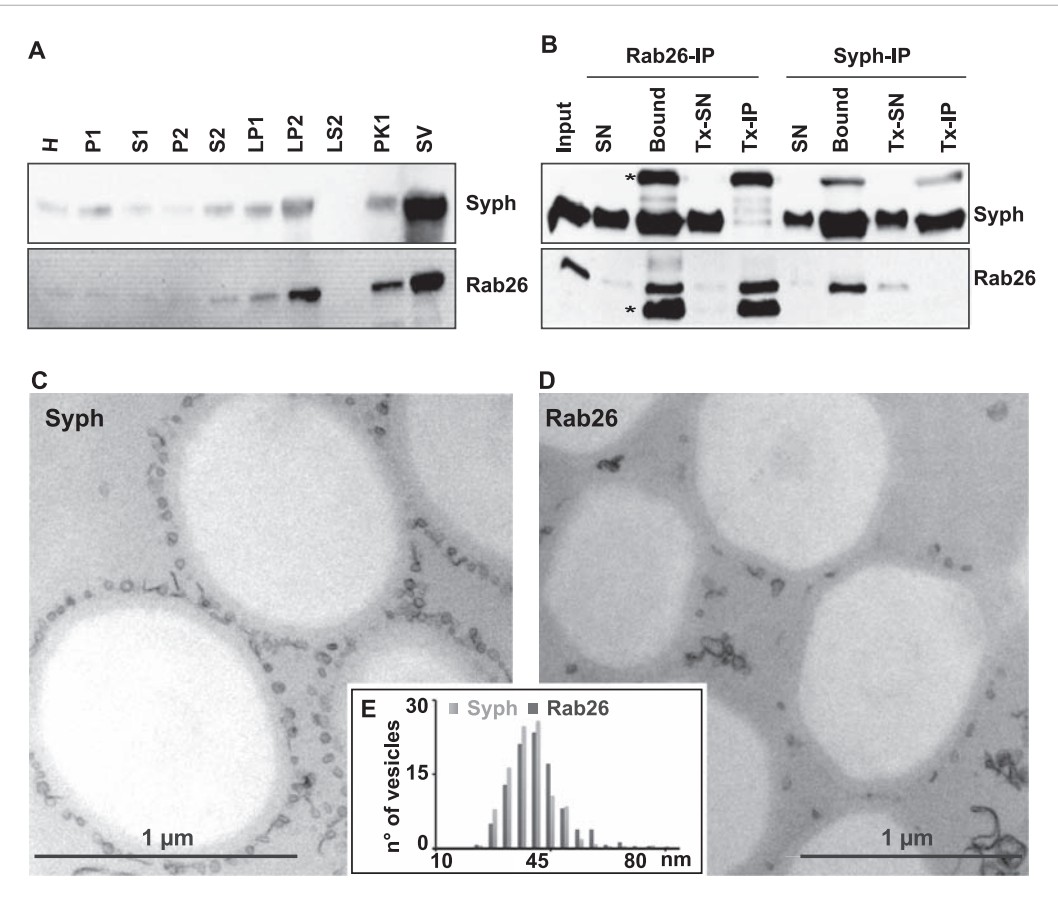

**Figure 1**. The small GTPase Rab26 co-purifies with synaptic vesicles. (**A**) Rab26 co-purifies with synaptic vesicles using conventional fractionation. Synaptic vesicles were purified from rat brain homogenate (H) by two consecutive differential centrifugation steps, yielding a low-speed pellet (P1) and a supernatant (S1), followed by a second centrifugation yielding a pellet P2 (containing synaptosomes and mitochondria) and a supernatant (S2). P2 was then lysed by osmotic shock, followed by centrifugation to separate large particles including synaptic junctional complexes (LP1) and a supernatant from which small membranes enriched in synaptic vesicles are collected by high-speed centrifugation (LP2, supernatant LS2 only contains soluble proteins). LP2 was further fractionated by sucrose density gradient centrifugation followed by chromatography on controlled pore glass beads where larger membrane fragments (PK1) were separated from synaptic vesicles (SV) (*Huttner et al., 1983*). 12 µg of proteins of each fraction was analyzed by SDS-PAGE and immunoblotting for Rab26 and the synaptic vesicle marker synaptophysin. Note that Rab26 copurifies with synaptophysin, displaying a pattern typical of synaptic vesicle proteins. (**B**) Synaptic vesicles were immunoisolated using Eupergit C1Z/methacrylate beads to which monoclonal antibodies specific for Rab26 or synaptophysin (Syph) respectively, were covalently coupled. The beads were incubated with a resuspended LP2 fraction and collected (see [*Burger et al., 1989*; *Takamori et al., 2000*] for details). SN, supernatant obtained after bead incubation; Bound, immunoisolates; Tx-SN and Tx-IP, same as before but the Input sample was solubilized with Triton X-100 to a final concentration of 1% before immunoprecipitation. For immunoblotting, 4% of the input or supernatant samples and 33% of bound samples were loaded for analyses. Note that Rab26 and synaptophysin cofractionate with the immunobeads irrespective of the antibody employed. Incubation with synaptophysin beads quantitatively depleted Rab26 from the supernatant whereas depletion of synaptophysin by Rab26-beads was less complete. In contrast, only the respective antigen was recovered from the detergent-solubilized samples. Asterisks denote IgG heavy and the light chains of the antibodies used for isolation, respectively. (**C** and **D**) Electron microscopy showing ultrathin sections of methacrylate beads containing bound organelles that were captured with synaptophysin- (**C**) or Rab26-specific (**D**) antibodies, respectively. (**E**) Size distribution (diameter) of bead-associated vesicles. Note that both populations exhibit a very similar and homogeneous size distribution, with a peak between 40–45 nm as is characteristic for synaptic vesicles (*Takamori et al., 2006*).

The following figure supplement is available for figure 1:

**Figure supplement 1**. Monoclonal Rab26 antibody generated in this study specifically recognizes Rab26.

Nevertheless, quantitative assessment of the size distribution of the bead-bound vesicles revealed no distinguishable difference between the vesicles bound to synaptophysin and Rab26 beads, respectively (*Figure 1E*).

Like other GTPases, Rabs function as molecular switches which undergo conformational changes during GDP/GTP cycling (*Grosshans et al., 2006*; *Stenmark, 2009*). This cycle is frequently paralleled by a membrane 'on/off' association—dissociation cycle. Membrane association is achieved following posttranslational modification of Rabs by geranyl-geranylation, a prerequisite for membrane insertion and Rab activation. Conversely, membrane dissociation is regulated by a specific adaptor protein, termed GDP dissociation inhibitor (GDI), which sequesters GDP-bound Rabs from membranes to the cytosol following GTP-hydrolysis (*Araki et al., 1990*; *Ullrich et al., 1993*; *Goody et al., 2005*; *Wu et al., 2010*). Therefore, to gain initial insights into the functional membrane association/dissociation cycle of Rab26, we assessed whether it could be extracted from synaptic vesicle membranes following GDI treatment. For this, we incubated a fraction enriched in synaptic vesicles (LP2) with purified recombinant GDI in the presence of GDP or GTPγS. The latter analogue was used to overcome the intrinsically high GTPase activity of Rabs (see 'Discussion'). Consistent with previous observations, Rab3 is rapidly retrieved from synaptic vesicles membranes by GDI in the presence of GDP (*Araki et al., 1990*; *Chou and Jahn, 2000*; *Pavlos et al., 2010*). By comparison, Rab26 is resistant to GDI-mediated membrane extraction, even when GDP is present in excess (*Figure 2A*). This feature is reminiscent of the biochemical characteristics of Rab27b, which also fails to be retrieved from synaptic vesicles by GDI treatment in vitro (*Pavlos et al., 2010*). Rather, Rab27b is known to dimerize and persist on synaptic vesicle membranes in its GDP-bound form (*Chavas et al., 2007*; *Pavlos et al., 2010*). Therefore, to assess whether GDP-bound Rab26 exhibits a similar tendency to oligomerize, we additionally performed co-immunoprecipitation experiments between FLAG- and EGFP-tagged wild-type (WT) Rab26 and its variants containing mutations that selectively interfere with the Rab26 GDP/GTP switch domain(s). As shown in *Figure 2B*, co-precipitation of FLAG-tagged Rab26 was only observed when cells were transfected with either wild-type EGFP-Rab26 or with a GDP-preferring variant (Rab26T77N, henceforth referred to as Rab26TN). By comparison, little to no co-precipitation was observable when the 'nucleotide empty' Rab26 (Rab26N177I, Rab26NI) or 'GTP-locked' (Rab26Q123L, Rab26QL) variants were employed. Together, these data indicate that Rab26 is a synaptic vesicle protein that oligomerizes preferentially in its GDP-bound form, thereby precluding GDI-mediated membrane extraction—a feature shared with its synaptic vesicle relative Rab27b.

Next, to study its subcellular localization in more detail, we immunostained primary cultures of rat hippocampal neurons for Rab26. First, the distribution of endogenous Rab26 was compared with that of synaptotagmin-I, one of the major membrane constituents of synaptic vesicles. As shown in *Figure 3A*, Rab26 labeling resulted in a conspicuous punctate staining pattern that overlapped with, although was not identical to, the pattern obtained with synaptotagmin-I antibodies. Higher magnification of neurites revealed that most of the Rab26 positive puncta colocalized with synaptotagmin-I. In contrast, many puncta positive for synaptotagmin-I were not stained with the Rab26 antibody (*Figure 3B*, arrows show colocalization).

To shed more light on the intracellular distribution of Rab26, neurons were transiently transfected with variants of FLAG-tagged Rab26 and then labeled for FLAG and synaptotagmin 1 (*Figure 3*). The staining pattern obtained with Rab26WT (*Figure 3C*) and with Rab26QL (the GTP-preferring variant) (*Figure 3D*) was very similar to that of endogenous Rab26, showing a high degree of overlap with endogenous synaptotagmin 1. In contrast, the GDP-preferring Rab26TN exhibited a more diffuse but still somewhat granular pattern that displayed no significant co-localization with synaptotagmin 1 (*Figure 3E*). A similar staining pattern was also observed when EGFP-tagged wildtype Rab26 was used, with the fusion protein again colocalizing with synaptic vesicle markers (in this case synaptophysin) (*Figure 3F*) and with the neuropeptide RFP-NPY but not with the early endosomal marker/Rab5 effector EEA1 (*Figure 3—figure supplement 1A,B*, respectively). This suggests that Rab26 is likely to traffic in a pathway distinct from Rab5.

Next, we tested whether the Rab26 positive puncta represent synapses undergoing exo-endocytosis. To monitor this, live hippocampal neuronal cultures expressing EGFP-Rab26 were pre-incubated overnight with a fluorescently-conjugated antibody specific for the luminal domain of synaptotagmin-I.

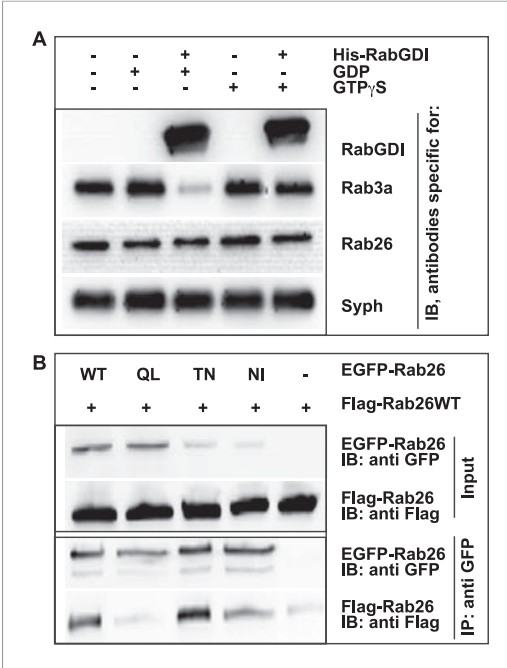

**Figure 2**. GDP-Rab26 cannot be extracted by GDI from membranes and forms oligomers. (**A**) Rab26 is resistant to extraction by GDI from synaptic vesicle membranes. An enriched synaptic vesicle fraction (LP2) was incubated with GTPγS or GDP (500 µM) for 15 min at 37°C. His-GDI (5 µM) or PBS (control, first lane) was added and the samples were incubated for an additional 45 min at 37°C. The membranes were then separated from the soluble fraction by centrifugation and analyzed by immunoblotting. While Rab3a was efficiently depleted from synaptic vesicles, Rab26 persisted on membranes. IB, immunoblotting. (**B**) Rab26 oligomerizes in a GDP-dependent manner. HEK293 cells transiently co-expressing EGFP-Rab26 variants (WT, QL, TN or NI) with FLAG-Rab26WT were lysed in detergent containing buffer followed by immunoprecipitation of EGFP-Rabs. Co-precipitation of FLAG-Rab26WT was observed with EGFP-Rab26 WT and even more efficiently with the GDP-preferring variant Rab26TN whereas co-precipitation with the nucleotide-empty variant (Rab26NI) was reduced and binding to the GTP-preferring variant (Rab26QL) was abolished. IP, immunoprecipitation.

This antibody can only bind to synaptotagmin when the luminal domain of the protein is exposed to the extracellular surface following synaptic vesicle exocytosis, and is thus used to conveniently identify active synapses as well as synaptic vesicles that have undergone exo-endocytosis (*Kraszewski et al., 1995*; *Fernandez-Alfonso et al., 2006*). As shown in *Figure 3G*, many of the EGFP-Rab26 puncta colocalized with vesicles labeled with this antibody, thereby confirming that these Rab26-positive vesicles originated from endocytosis of vesicles that previously had undergone at least one round of exocytosis.

Rab26 is one of the Rab GTPases conserved between mammals and *Drosophila*, with the genome of latter encoding three alternatively spliced isoforms. According to a systematic analysis of all *Drosophila* Rab proteins, Rab26 is expressed specifically in neurons at all developmental stages (larval and pupal development, adults flies) (*Chan et al., 2011*). To test whether the distribution of Rab26 in *Drosophila* resembles that in cultured hippocampal neurons, we expressed YFP-tagged versions of wild-type (WT), GTP-preferring (RabQ250L) and GDP-preferring (Rab26T204N) Rab26 using the pan neuronal *elav*-Gal4 driver (*Zhang et al., 2007*). Although the tagged Rab26 protein variants were expressed throughout development, no lethality or delayed development was observed (data not shown). Consistent with previous observations, analysis of third instar larvae nerve-muscle preparations revealed an exclusive localization of Rab26 to presynaptic compartments of the neuromuscular junction without staining of axons and cell bodies (*Figure 3—figure supplement 2*; see also [*Chan et al., 2011*]). Remarkably, expression of wild-type and of the gain-of-function (QL) Rab26 resulted in the appearance of large vesicle clusters within the presynaptic boutons whereas the GDP-preferring form (TN) was diffusely localized. This indicated that, like in hippocampal neurons, formation of these clusters is dependent on the nucleotide-bound state of Rab26 (*Figure 3H*). These Rab26-positive compartments are present in neuromuscular junction boutons as indicated by staining with anti-horseradish peroxidase (HRP) (*Figure 3—figure supplement 2*) and showed a partial overlap with the synaptic vesicle protein cysteine string protein (Csp) (*Figure 3H*) (*Zinsmaier et al., 1990*). Conspicuously, large Rab26-positive structures often showed intense Csp staining at their borders. Importantly, Rab26 is excluded from active zones immunostained for the Bruchpilot (Brp), a scaffold protein specifically localized to active zones (*Figure 3—figure supplement 2*) (*Wagh et al., 2006*).

## Rab26 is associated with pre-autophagosomal compartments

Whereas the distribution of endogenous and tagged-expressing Rab26 variants was comparable in neurites, expression of EGFP-Rab26WT in cultured hippocampal neurons resulted in a unique and highly conspicuous phenotype that was not observed with endogenous Rab26, untagged or

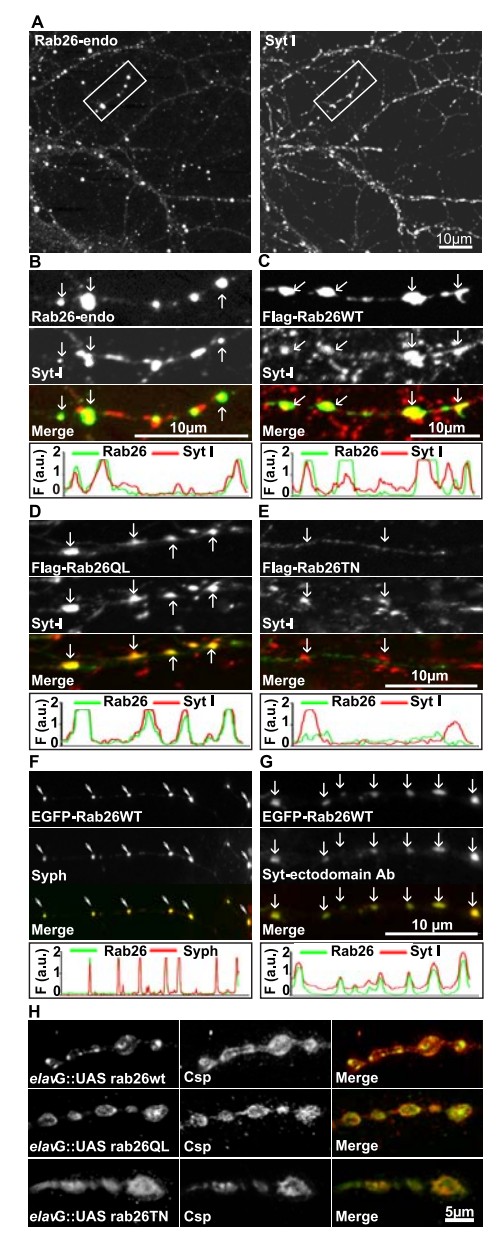

**Figure 3**. Endogenous and expressed GTP-Rab26 variants localize to a subset of synaptic vesicles in cultured hippocampal neurons. In (**A–G**), representative line scans of the two channels are shown below each set. In the y-axis, F (a.u.) indicates fluorescence intensity (arbitrary units). (**A** and **B**) Localization of endogenous Rab26 (detected with the newly generated monoclonal anti Rab26 antibody) and synaptotagmin-I (Syt-I) in neurites of dissociated hippocampal neurons (DIV 15) reveals that Rab26 colocalizes with a subset of Syt-I positive puncta (**B**, arrows). (**C–E**) Expression of FLAG-tagged Rab26 variants in neurites (DIV 9 cultures 48hr after transfection). Both FLAG-Rab26WT (**C**) and QL (**D**) co-localize with a subset of synaptotagmin positive puncta (Syt-I), whereas FLAG-Rab26TN (**E**) exhibited

FLAG-tagged Rab26 and that has not been previously reported for any other EGFP-tagged Rab GTPase. In the soma, EGFP-Rab26WT induced the formation of large vesicular structures that, in some instances, filled the major part of the neuronal cytoplasm (*Figure 4*). These structures were intensely positive for both synaptic vesicle (*Figure 4A,B*) and large dense core vesicle markers, that is, the two types of neuronal secretory vesicles that usually do not show significant overlap (*Figure 4C* and *Figure 4—figure supplement 1C*). Intriguingly, the overlap with Rab3A, the major Rab-GTPase associated with synaptic vesicles, was less apparent. Double labeling with a variety of compartment-specific markers revealed no overlap with early endosomes or Golgi (*Figure 4—figure supplement 1A,B*, respectively) but some, albeit limited, overlap with lysosomes (*Figure 4D*).

To better understand the nature of these structures we carried out immunogold-TEM on ultrathin cryosections of transfected neurons. As shown in *Figure 5A*, the soma contained large clusters of numerous small vesicles that were positive for both EGFP-Rab26WT and synaptobrevin (Sybv). In many cases, these clusters were rather homogenous, but occasionally also contained larger vesicles and mitochondria (*Figure 5B*). Although no systematic quantification was performed, some of these clusters reached enormous dimensions containing possibly 1000s of small vesicles (*Figure 5—figure supplement 1*). In some cases, these clusters were surrounded by a single and/or double membrane (*Figure 5C*, inset), although this was somewhat variable. We also assessed neuromuscular junctions of *Drosophila* strains overexpressing YFP-Rab26WT by TEM. Here, dense clusters of vesicles (devoid of surrounding membranes) were regularly observable that were clearly set apart from surrounding synaptic vesicles (*Figure 5D*, indicated by an arrow) but were clearly absent in controls.

As described above, in neurites Rab26 is associated with large clusters containing synaptic vesicle proteins regardless of whether endogenous Rab26 is visualized or whether Rab26 is overexpressed. Thus, it is conceivable that the induction of these vesicle clusters is an intrinsic property of GTP-Rab26, which is enhanced by the weak homo-dimerization of EGFP and YFP (*Shaner et al., 2005*). What could be the identity of these clusters? Considering their striking resemblance of these clusters to autophagosomal precursors, we hypothesized that these clusters may represent autophagosomes at various stages of formation and/or maturation.

*Figure 3. Continued*

a more diffuse distribution. (**F**) Overexpression of EGFP-Rab26WT exhibits a distribution comparable to endogenous and FLAG-tagged Rab26 in neuritis where it colocalizes with synaptophysin (Syph). (**G**) Rab26-positive clusters contain actively recycling vesicles. Hippocampal cultures were incubated overnight with a labeled antibody directed against a lumenal epitope of synaptotagmin (Syt-I clone 604.2 conjugated to Oyster-550), resulting in uptake during endocytosis. EGFP-Rab26WT is present in recycled synaptic vesicles as evident from the co-localization with synaptotagmin-I positive puncta. (**H**) Localization of YFP-tagged Rab26 variants at the Drosophila neuromuscular junction (third instar larvae). The Rab26 variants (Rab26wt, GTP-preferring Rab26Q250L, and GDP-preferring Rab26T204N), were expressed using elav-Gal4. Neuromuscular junctions of third instar larvae expressing these Rab26 variants were stained with anti-GFP (green) and for endogenous cysteine string protein as vesicle marker (anti-Csp, red).

The following figure supplements are available for figure 3:

**Figure supplement 1**. EGFP-Rab26WT colocalizes with the secretory neuropeptide Y (NPY) but not with EEA1 in neurites.

**Figure supplement 2**. Rab26 does not colocalize with the presynaptic scaffold protein Brp in Drosophila neuromuscular junctions.

Autophagy is a degradative pathway during which cellular contents are enclosed by a double-membrane (i.e., the isolation membrane) and then delivered to lysosomes for disposal (for review see e.g., [*Mizushima et al., 2011*; *Lamb et al., 2013*]). The pathway is initiated by two ubiquitin-like conjugation systems that operate in a sequential manner. The first conjugates the ubiquitin-like protein Atg12 to Atg5 which is then recruited by Atg16L1 to the pre-autophagosome structure. This complex then recruits a LC3 family member, a second ubiquitin-like molecule, and attaches it covalently to phosphatidylethanolamine in an E3-ligase like reaction (*Klionsky and Schulman, 2014*). Since LC3 remains associated with the autophagosomal membrane until its delivery to the lysosome, it is considered to be the most reliable marker for autophagosomes (*Klionsky et al., 2012*).

Therefore, to test whether the Rab26 containing clusters are linked to the autophagy pathway, we next checked for association with autophagosome-related proteins. First, we assessed for colocalization between Rab26 and Atg16L1, a component of pre-autophagosomes (*Mizushima et al., 2003*; *Ravikumar et al., 2010*). For this purpose, hippocampal neurons transiently expressing EGFP-Rab26WT were immunostained for endogenous Atg16L1. Indeed, an almost perfect colocalization between Atg16L1 and Rab26-positive clusters was detected in neuronal cell bodies (*Figure 6A*), thereby identifying these

clusters as autophagosomal precursors. Next, we stained untransfected neurons for endogenous Rab26 and Atg16L1. Again, a high degree of overlap was observed between Rab26 and Atg16L1 in clusters decorating neurites (*Figure 6B*) but not in cell bodies which remained largely unstained (not shown). Strong overlap with Atg16L1 was also observed when neurons were transfected with FLAG-Rab26WT and/or FLAG-Rab26QL, but not with FLAG-Rab26TN (*Figure 6C*). This indicated that the association of Rab26 with autophagosomes depends on the GTP-form of the protein. This GTP-dependency was similarly noted in HeLa cells following ectopic expression of Rab26. In this instance, overexpression of GTP-bound forms (WT and QL), but not GDP-bound (TN) form, of EGFP-Rab26 led to the formation of large Atg16L1-positive clusters (*Figure 6—figure supplement 1A–C*). Analysis by immunogold-TEM again revealed that these clusters consisted of small but often heterogeneous vesicles, partially surrounded by membranes, with EGFP labeling detected both on vesicles within clusters as well as on their encapsulating membrane(s) (*Figure 6—figure supplement 1D*).

Transient association of Atg16L1 to pre-autophagosomal structures enables the recruitment and membrane attachment of LC3 family members that persist on the autophagosomal membranes until degradation. Therefore, to assess whether the Rab26/Atg16L1 clusters recruit LC3 to membranes, we co-transfected cultured hippocampal neurons with GFP-tagged LC3B (one of the eight known LC3 family members) and FLAG-tagged Rab26 variants. Co-expression of GFP-LC3 and active forms FLAG-Rab26 (WT and QL) resulted in a localization pattern comparable to that observed for Atg16L1 (*Figure 6D*), thereby verifying the nature of these compartments as autophagosomes. Interestingly, however, LC3 recruitment was not observed when the EGFP-tagged Rab26WT was overexpressed (not shown, see 'Discussion').

Taken together, the above findings indicate a novel functional link between a Rab GTPase and a hitherto unknown autophagy pathway in neurons that appears selective for synaptic and/or secretory

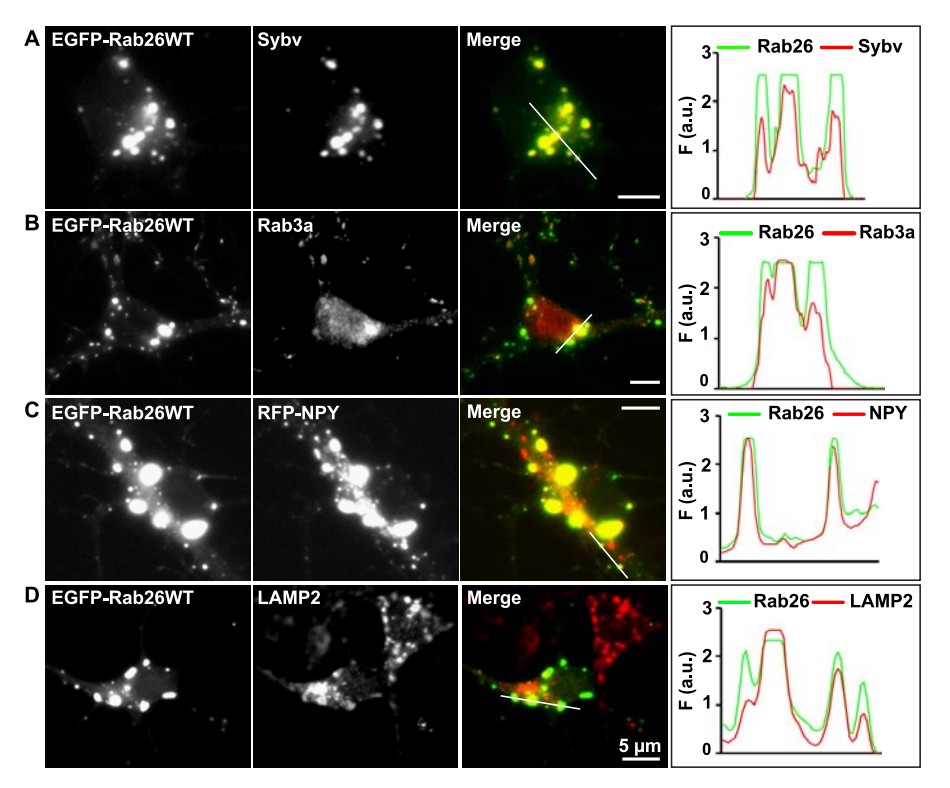

**Figure 4**. Expression of EGFP-Rab26WT in cultured hippocampal neurons induces formation of large GFP-positive clusters in the neuronal cell bodies. (**A–C**): GFP-positive clusters colocalize with markers for synaptic and large dense core vesicles. (**A** and **B**) EGFP-Rab26WT clusters contain synaptobrevin (Sybv) and Rab3a. (**C**) Co-expression of EGFP-Rab26WT with RFP-NPY results in almost complete overlap of both proteins in these clusters (**C**). (**D**) Partial overlap is also observable with the lysosomal membrane protein LAMP2. DIV 10, scale bar, 5 μm. Line scans of select vesicle clusters (denoted by solid white lines in merged panel) signify the relative correlation between the individual fluorescent channels. F (a.u.) indicates fluorescence intensity (arbitrary units).

The following figure supplement is available for figure 4:

**Figure supplement 1**. EGFP-Rab26WT colocalizes with the secretory protein secretogranin II but not with EEA1 and GM130 in neuronal somata.

vesicles, tentatively termed 'vesiculophagy'. Indeed, there is now a growing body evidence implicating several Rabs (Rab1, Rab7, Rab9, Rab11, Rab24, Rab32, and Rab33, inclusive) in canonical autophagy (for a comprehensive review [*Chua et al., 2011*]). Among these, Rab33, a Golgi resident Rab, participates in the formation of autophagosome precursors by recruiting Atg16L1 (a Rab33 effector) to isolation membranes (*Itoh et al., 2008*). Given that Rab26 colocalizes with Atg16L1, we checked for potential cooperation between Rab26 and Rab33 in neurons. For this, hippocampal neurons were co-transfected with FLAG-Rab26WT and EGFP-Rab33BWT. As shown in *Figure 6E*, EGFP-Rab33BWT was largely restricted to the perinuclear/Golgi region in the soma which showed no appreciable overlap with FLAG-Rab26WT. On the other hand, significant overlap between Rab26 and Rab33 was observed in more peripheral puncta lining neurites (*Figure 6F*). Together these data imply that the autophagy-pathway regulated by Rab26 may functionally intersect with Rab33.

## Atg16L1 is an effector of Rab26

The overlap between Rab26 and Rab33 prompted us to further investigate whether Atg16L1 may also be an effector of Rab26. To explore this possibility, we performed co-immunoprepitation experiments

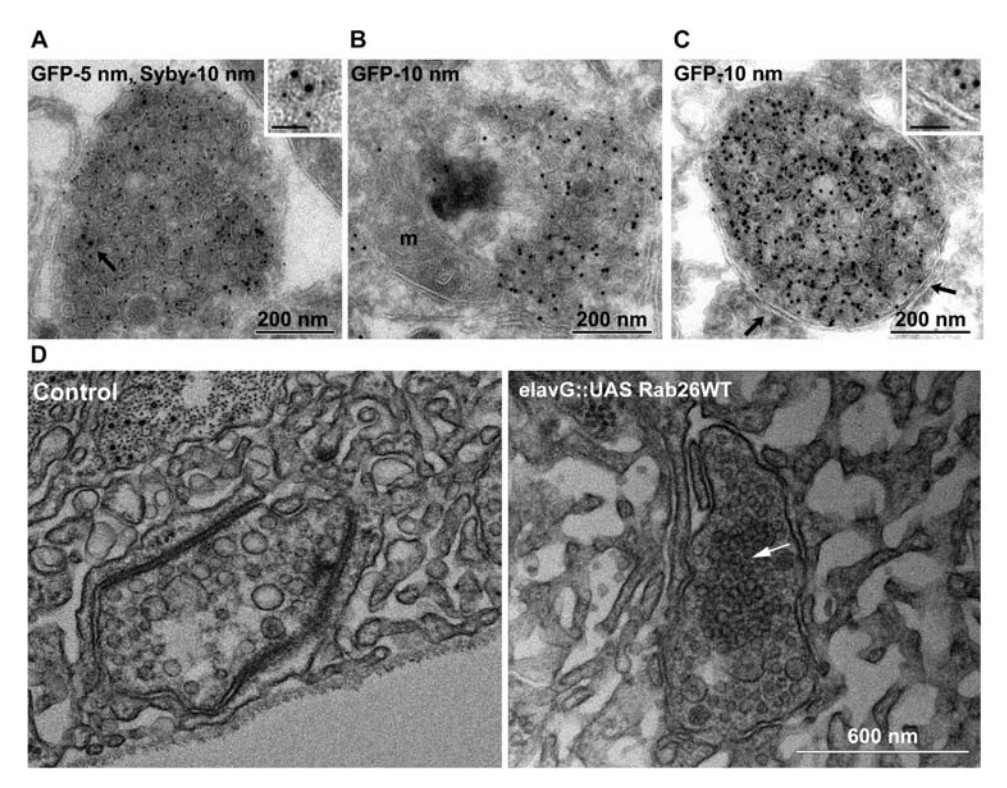

**Figure 5**. Ultrastructure of EGFP-Rab26 induced vesicle clusters in cultured hippocampal neurons and in neuromuscular junctions of Drosophila third instar larvae. (**A–C**) Ultrathin cryosections obtained from hippocampal neurons expressing EGFP-Rab26WT were immunogold labeled for EGFP and synaptobrevin (Sybv, monoclonal antibody 69.1 specific for synaptobrevin 2) and analyzed by electron microscopy. In the soma of hippocampal neurons organelles surrounded by one or two (**C**, inset, arrow) membranes were densely packed with small vesicles and very occasionally other organelles (e.g., mitochondria, m). Immunogold labeling for both EGFP and synaptobrevin was concentrated both on vesicles present inside and the surrounding membrane. Scale bar in insert, 50 nm. (**D**) Ultrathin sections of neuromuscular junctions obtained from Drosophila third instar larvae. Control animals show scattered vesicles of a somewhat heterogeneous size, typical for this developmental stage (*Rasse et al., 2005*). Synapses of a strain expressing YFP-Rab26WT using the elavG::UAS system (elavG::UAS Rab26WT) display frequent clusters of densely packed vesicles (arrow) that are separated from the surrounding vesicles but lack a surrounding membrane.

The following figure supplement is available for figure 5:

**Figure supplement 1**. EGFP-Rab26WT induces massive vesicle clustering in neuronal cell bodies.

between FLAG-tagged Rab26 (WT, QL or TN) and endogenous Atg16L1 in HeLa cells. As shown in *Figure 7A*, all three FLAG-tagged Rab26 variants were efficiently immunoprecipitated with the FLAG antibody. Immunoblotting for endogenous Atg16L1 from the same immunoprecipitates revealed co-precipitation between Atg16L1 and Rab26QL. By comparison, little to no Atg16L1 was detectable in the precipitates of Rab26-WT and Rab26TN, respectively, indicating that the interaction between Rab26 and Atg16L1 is GTP-dependent.

In parallel, we performed GST-pulldown assays to verify the results from the coIP experiments. For this, purified bacterially expressed recombinant Rab26 variants (QL or TN), tagged with GST were incubated with a preassembled complex of Atg5 and the N-terminal fragment of Atg16L1 containing its coiled coil domain (Atg16NT). In agreement with our immunoprecipitation studies, GST-pulldown revealed an interaction between Atg16L1 and Rab26, with Atg16L1 binding to the QL and to a lesser extent the TN-variant of Rab26 (*Figure 7B*), with the latter being further reduced upon repetitive washings (not shown).

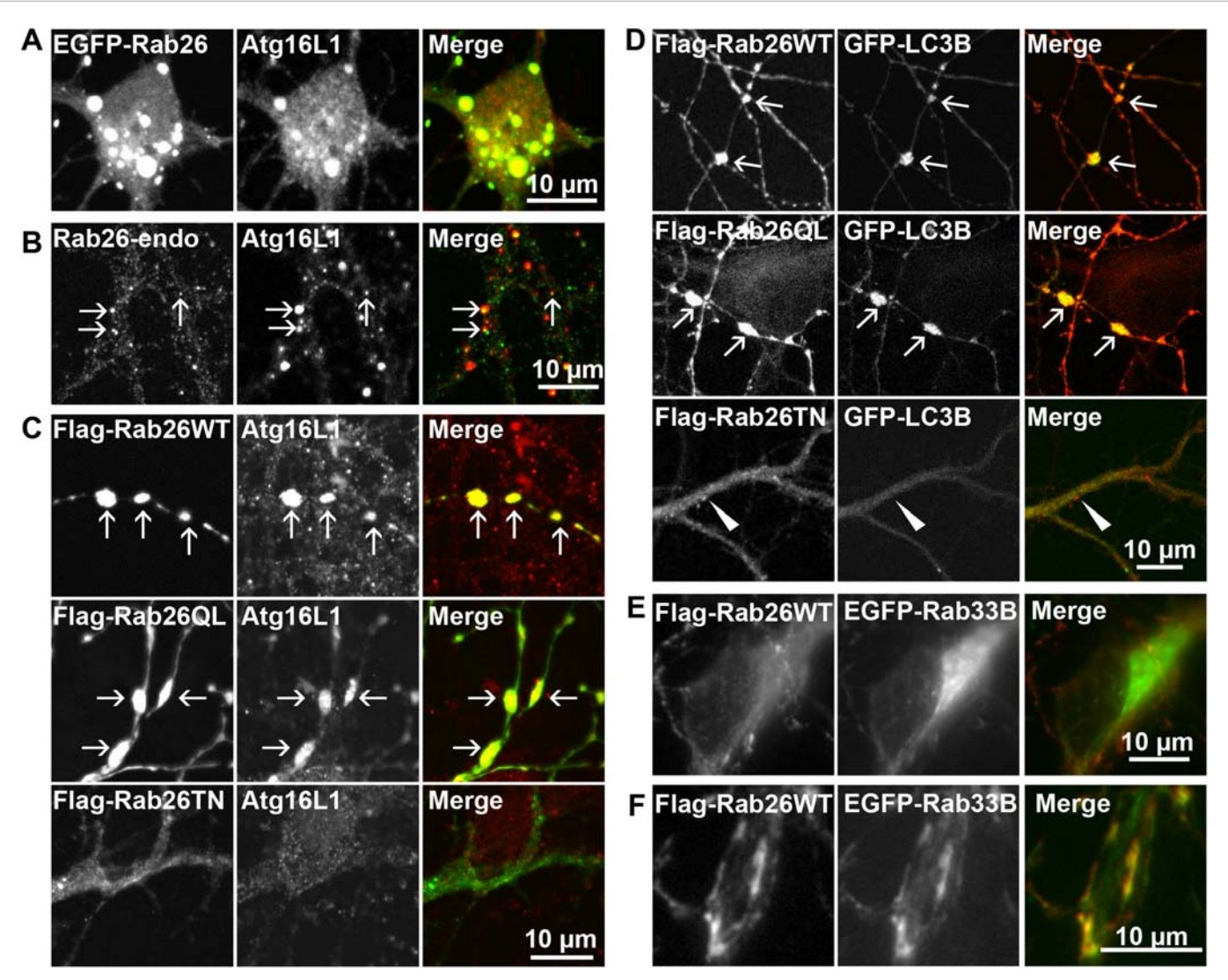

**Figure 6**. Clusters containing GTP-Rab26 colocalize with autophagosome-specific proteins both in cell bodies and dendrites of cultured hippocampal neurons. Arrows indicate co-localization. (**A**) Somatic clusters induced by expression of EGFP-Rab26WT co-localize with endogenous Atg16L1. (**B** and **C**) In neurites, Atg16L1 co-localizes with clusters of endogenous Rab26 (non-transfected, DIV 15, panel **B**) and with clusters containing FLAG-tagged Rab26WT and Rab26QL, but not with Rab26TN (transfected, DIV 9, panel **C**). (**D**) Similar colocalization patterns were obtained from neurites expressing FLAG-Rab26 variants and autophagosomes labeled by GFP-LC3B. Note that occasional puncta were observed for the GDP-preferring variant Rab26TN that, however, showed no overlap with LC3B (arrowhead). DIV 9. (**E** and **F**) Co-expression of FLAG-Rab26WT and EGFP-Rab33WT in hippocampal neurons. In the soma (**E**), EGFP-Rab33 in primary restricted to a perinuclear structure reminiscent of the Golgi apparatus whereas an almost perfect overlap was observed between the Rab33 and Rab26 in peripheral puncta lining neurites (**F**). DIV 8.

The following figure supplement is available for figure 6:

**Figure supplement 1**. Rab26 overexpression causes vesicle clusters in HeLa.

Atg5 remains bound in this complex. To further examine the interaction, we analyzed the binding between Rab26 and Atg16L1 using analytical gel filtration. Surprisingly, formation of Rab26(QL)-ATG16L1 complexes were not detectable with this approach (***Figure 7—figure supplement 1***). As a positive control, we carried out the same experiment using Rab33(QL) and ATG16L1. Here, complex formation was detectable with this approach. Thus, while both IP and pull-down experiments show that RAB26 binds ATG16L1 in a GTP-dependent manner, this binding appears to be weaker than the interaction between Rab33 and ATG16L1.

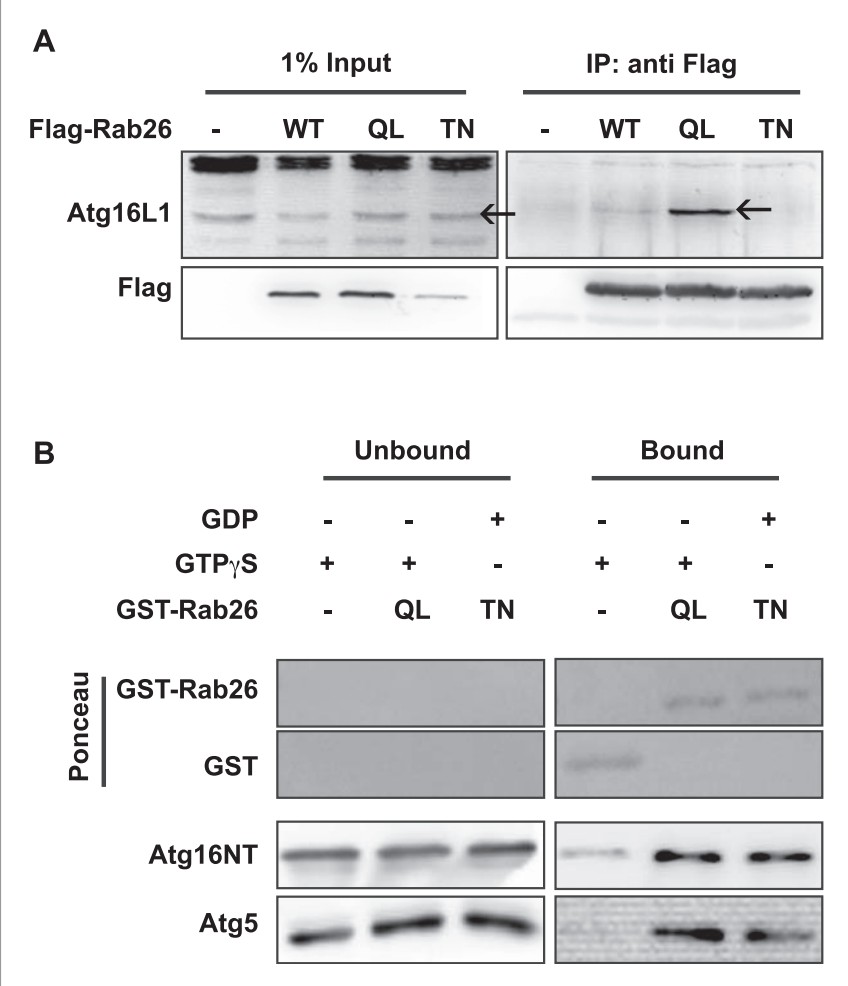

**Figure 7**. Atg16 is an effector of GTP-Rab26. (**A**) Co-Immunoprecipitation of FLAG-tagged Rab26 variants expressed in HeLa cells with endogenous Atg16L1 protein. Immunoprecipitation was carried out following lysis in detergent-containing buffer and clearance by centrifugation to remove cell debris. Note that only the GTP-preferring QL variant of Rab26 showed significant binding to Atg16L1 (arrow). (**B**) GST pulldown of purified recombinantly expressed GST-Rab26 variants with a pre-formed complex of His-tagged versions of Atg5 and the N-terminal domain of Atg16L1 (Atg16NT). Note that Atg16NT selectively interacted with the GTP-preferring QL-variant of Rab26.

The following figure supplement is available for figure 7:

**Figure supplement 1**. Analyses of Rab26 and ATG16L1 interaction by analytical gel filtration.

## Discussion

In the present study we have combined multiple complementary biochemical and cell biological approaches to demonstrate that the small GTPase Rab26 is specifically associated with synaptic vesicles. Intriguingly, Rab26 appears to be particularly enriched in large clusters of synaptic vesicles to which the autophagy proteins Atg16L1, LC3 and Rab33B are recruited, suggesting that they represent pre-autophagosomal compartments. We show further that, at least when using overexpression of EGFP-tagged Rab26, such clusters are also formed in cell bodies where they are enclosed by a single (and in some instances a double) isolation membrane. Based on these observations, we conclude that Rab26 provides a direct link between synaptic vesicles and the core autophagy machinery thereby uncovering a novel and selective autophagy pathway (tentatively named 'vesiculophagy') that may regulate synapse turnover.

## Rab26 is a secretory GTPase resembling Rab27b in its molecular properties but differing in its function

Rab26 is most closely related to the secretory GTPases Rab3 and Rab27, which led to the conclusion that it may perform similar functions in membrane traffic (*Fukuda, 2008*). This view is supported by reports showing association of Rab26 with zymogen granules in exocrine cells (*Nashida et al., 2006*) and by the observation that overexpression of the dominant-negative form abolishes the formation of zymogen granules (*Tian et al., 2010*; *Li et al., 2012*). More recently, Rab26 has been found to be associated with lysosomes in zymogen-secreting cells (*Jin and Mills, 2014*) implying that its functions in secretory cells extend beyond that of exocytosis. In our previous work (*Takamori et al., 2006*; *Pavlos et al., 2010*), Rab26 was identified among the list of Rab proteins (up-to-30) enriched on highly purified synaptic vesicles. Our present data now show that this association is exclusive, with Rab26 being absent from other organelles such as early endosomes, paralleling the distribution of other secretory Rabs. On the other hand, the preferential association of Rab26 with large clusters of synaptic vesicles and its conspicuous absence from smaller boutons positive for synaptic vesicle markers is clearly distinct from Rab3 and Rab27b and indicates that Rab26 may not be contributing to the canonical function of these Rabs in regulated exocytosis.

Intriguingly, in contrast to for example, Rab3 and Rab5, Rab26 cannot be extracted from synaptic vesicle membranes by GDI in its GDP-form—a feature it shares with Rab27b. Rather, Rab26 exhibits a tendency to oligomerize in the GDP-form, again a feature shared with Rab27b and perhaps with some others such as Rab11 and Rab9, which crystallize as dimers in the GDP-state (*Pasqualato et al., 2004*; *Wittmann and Rudolph, 2004*; *Chavas et al., 2007*). It is somewhat surprising that, along with the GDP-bound variant, wild-type Rab26 also appears to oligomerize (albeit to a lesser extent). However, this might be explained by the known high intrinsic GTP-hydrolysis rates of wild-type and native Rab proteins which would render a percentage of wild-type Rab26 inactive (approximately 40% by co-immunoprecipitation), thereby prompting its oligomerization on membranes and hence preclude its GDI-extraction from synaptic vesicles. By comparison, the propensity of the 'GDP-locked' variant (Rab26TN) to rapidly oligomerize would likely result a steric clash with the GDI-related Rab escort protein (REP), which is required for prenylation and delivery of new synthesized Rabs to their cognate membranes (*Andres et al., 1993*). Such a scenario would be in keeping with the predominantly cytosolic distribution pattern observed in neurons expressing FLAG/EGFP-tagged Rab26TN.

Perhaps the most conspicuous feature of Rab26 is that it is not only preferentially associated with secretory vesicle clusters but actually induces their formation in a GTP-dependent manner as becomes apparent upon the expression of exogenous Rab26 variants in both neurons and non-neuronal cells. This is most dramatically observed with the EGFP-tagged variant suggesting that the weak homodimerization tendency of EGFP enhances the phenotype (note that no other EGFP-tagged Rab exhibits similar features including the most abundant secretory GTPase, Rab3a). At present, the exact mechanism underlying this clustering phenotype is unclear. Nevertheless, since the GTP-form of Rab26 does not oligomerize, it is unlikely that clustering is effected by homophilic Rab26 interactions. Rather, it possible that clustering is mediated by a hitherto unknown effector protein. This effector is probably distinct from Atg16L1 as overexpression of EGFP-Rab33B (that also recruits Atg16L1) does not induce such clusters (*Figure 6*, and data not shown). However, given that the central terminal region of Atg16L1 has a tendency for homo-multimerization, this possibility cannot be excluded (*Mizushima et al., 2003*; *Ishibashi et al., 2011*). Intriguingly, our findings agree with a recent report according to which overexpression of Rab26 in exocrine cell lines induces clustering of lysosomes, reminiscent of the partial co-localization of the EGFP-induced Rab26 clusters with lysosomes in neuronal cell bodies (*Jin and Mills, 2014*).

## Does Rab26 direct synaptic vesicles towards the autophagy pathway?

Our results indicate that the core autophagy protein Atg16L1 is an effector of Rab26 that binds to the GTPase exclusively in the GTP-form, paralleling previous findings on the Golgi-resident Rab33B (*Itoh et al., 2008*). Interestingly, binding of Rab26 to Atg16L1 appears to be weaker than that between Rab33 and Atg16L1, which plays a role in canonical autophagy, probably explaining why *Itoh et al. (2008)* did not observe binding of GST-Rab26 to a full-length Atg16L1 protein using stringent conditions. It is conceivable that the interaction is more transient, or else, that it requires additional factors for stabilization, thus allowing for fine-tuning the flow of synaptic vesicles targeted for selective autophagy.

How does recruitment of Atg16L1 to synaptic vesicle clusters relate to the established steps of autophagosome formation? First of all, it cannot yet be excluded with certainty that upon recruitment to these vesicles Atg16L1 performs a non-canonical function that is not related to autophagosome formation (see e.g., [*Pimentel-Muinos and Boada-Romero, 2014*] for review of such functions). In particular, Atg16L1 and Rab33A have recently been found to be associated with secretory vesicles in neuroendocrine PC12 cells, with the data suggesting a role for Atg16L1 in regulating exocytosis independent of autophagy (*Ishibashi et al., 2012*). On the other hand, based on our extensive morphological assessment using double immunolabeling microscopy, we strongly favor that the Rab26-Atg16L1 complexes in neurons represent pre-autophagosomal structures because (i) Rab26 is not present on all synaptic vesicles but rather confined to vesicle aggregates that may be functionally impaired, and (ii) LC3 is recruited to these clusters suggesting that the formation of an autophagosomal membrane is, at least in part, initiated.

Our data indicates that the vesicle clusters containing Rab26 and Atg16L1 have undergone exo-endocytotic cycling. Intriguingly, clathrin has recently been shown to interact with Atg16L1, thus targeting plasma membrane constituents towards autophagosome precursors via clathrin-mediated endocytosis (*Ravikumar et al., 2010*). Since clathrin-mediated endocytosis constitutes the main endocytotic pathway for synaptic vesicles, it is conceivable that there is a synergy between Rab26- and clathrin-induced autophagocytosis in nerve terminals that further fine-tunes the targeting of synaptic vesicles to preautophagosomal structures.

Taken together, the data support the view that Rab26 is member of a signaling cascade that selectively targets synaptic and secretory vesicles towards autophagocytosis, which may represent a novel and highly specific form of autophagy ('vesiculophagy'). In recent years, it has become apparent that in addition to the classical, mTOR/ULK kinase-induced pathway of macroautophagy there is a panoply of diverse and highly regulated pathways (e.g., pexophagy and mitophagy) that all converge on the common pathway but differ in the mechanism of initiation and cargo recruitment. In many of these cases the pathway is initiated by ubiquitination of target proteins. While we do not know whether this is also the case here, it is conceivable that the initiation event may indeed be the recruitment of active Rab26 to the membrane of subsets of synaptic vesicles that then interacts with other factors to form clusters and to recruit an isolation membrane, the origin of which remains to be identified.

## Implications for presynaptic function

Following the classical work in the early 70s of last century demonstrating that synaptic vesicles undergo multiple rounds of recycling in the synapse, (*Atwood et al., 1972*; *Ceccarelli et al., 1973*; *Heuser and Reese, 1973*), recent attention has focused primarily on unraveling the mechanisms of endocytosis and vesicle re-formation (*Saheki and De Camilli, 2012*). However, all membrane constituents age and accumulate structural defects requiring sorting out of damaged constituents. Although no increase in the number of late endosomes, lysosomes or autophagosomes was observed following even massive stimulation, it was hypothesized as early as 1971 that newly reformed synaptic vesicles could either be actively re-used as functional synaptic vesicles or re-directed to a pathway ultimately leading to lysosomes as the final destination for degradation (*Holtzman et al., 1971*). Our discovery of vesiculophagy as a pathway initiated in presynaptic boutons that directs entire synaptic vesicle pools towards autophagosomes provides a previously uncharacterized link towards lysosomal degradation of trafficking organelles which is distinct from the classical endosomal route. Indeed, recent data suggest that presynaptic neurotransmission is functionally modulated by macroautophagy. Induction of autophagy in neurons increased the amount of autophagic vacuoles in presynaptic terminals and with an accompanying reduction in synaptic vesicle number and decrease in evoked neurotransmitter release (*Hernandez et al., 2012*). Furthermore, two groups have recently suggested that in axons autophagosomes originate distally and then are transported by retrograde motors towards the cell body. During their travel they undergo fusion with acidic compartments and finally with the lysosomes (*Lee et al., 2011*; *Maday et al., 2012*). It is therefore conceivable that Rab26 feeds vesicle membranes into autophagosomes that may form and mature during retrograde transport. How this novel pathway is initiated and regulated will be the subject of future studies.

# Material and methods

## Antibodies

Mouse monoclonal and rabbit polyclonal antibodies specific for synaptophysin, synaptotagmin, synaptobrevin, Rab3a, GDI (Cl 81.2) and GFP were provided by Synaptic Systems (Göttingen, Germany). Mouse anti-LAMP2 antibody was purchased from the Developmental Studies Hybridoma Bank (DSHB, University of Iowa, IA). Antibodies against EEA1 and GM130 were purchased from BD Bioscience (Franklin Lakes, NJ). The antibody against the FLAG epitope was obtained from Stratagene (La Jolla, CA). Antibodies specific for Atg16L1 were purchased from CosmoBio (Tokyo) and MBL (Nagoya). Anti-Atg5 antibody was from Novus Biological (Littleton, Colorado). The antibody against secretogranin II was kindly provided by Sharon Tooze (Cancer Research UK). The monoclonal antibody against Rab26 used in this study was raised by immunizing 8- to 10-week-old female BALB/c mice over a period of 17 days with full length recombinant human Rab 26. Cells from knee lymph nodes were fused with the mouse myeloma cell line P3X63Ag.653 (ATCC CRL-1580). Cell culture supernatants obtained from individual clones were then screened using enzyme-linked immunosorbent assay (ELISA), immunoblot assays and immunoflourescence. The final hybridoma used in this study was cloned two times by limiting dilution. The monoclonal antibody produced from this clone was determined to be of the IgG2a subclass and is specific for Rab26 (*Figure 1—figure supplement 1*). The antibody is commercially available from Synaptic Systems (Göttingen, Germany). Cy3-labeled goat anti-mouse or anti-rabbit and Alexa 488-labeled goat anti-mouse secondary antibodies were purchased from Dianova (Hamburg, Germany) and used at a dilution of 1:400. Horseradish peroxidase-conjugated anti-mouse and anti-rabbit secondary antibodies were obtained from Bio-Rad (Hercules, CA) and used at a dilution of 1:2000 or 1:4000.

## Plasmids

The coding sequence of human Rab26 (NM_014353.4) was amplified by PCR and inserted into pEGFP-C1 (Clontech, Palo Alto, CA) using BglII and BamHI restriction sites or into pCMV-Tag2a (Agilent Technologies, La Jolla, CA) using BamHI and XhoI restriction sites for expression in mammalian cells. Likewise, inserts encoding Rab26 Q123L, T77N or N177I mutants were generated by recombinant PCR and similarly inserted into these vectors. For recombinant protein expression in bacteria, inserts for the Rab26 variants were inserted into pGEX-KG using EcoRI and BamHI while the insert encoding alpha-GDI was sub-cloned into pET-28a (Novagen, Madison, WI). The sequence corresponding to murine Atg16L1(1–265) (BC049122) was cloned into pET-28a (Novagen) using NdeI and NotI restriction sites. Full-length murine Atg5(1–275) (BC002166) was cloned with an N-terminal thrombin cleavage site into the multiple cloning site 1 of pETDuet-1 (Novagen) using the SalI and NotI sites. The vector expressing neuropeptide Y (NPY) was generated by inserting the sequence encoding human pro-NPY into the pmRFP vector. Cloning was performed according to standard procedures (*Janssen, 2001*). The plasmid expressing GFP-tagged human LC3B was a kind gift from Dr Zvulun Elazar (Weizmann Institute, Israel).

## Cell cultures, transfection and immunocytochemistry

Culturing of the HEK 293 and HeLa SS6 cell lines and the preparation of high density primary rat hippocampal neurons have been previously described (*Rosenmund and Stevens, 1997*; *Chua et al., 2012*). Neurons were transfected between 7 to 12 days after seeding or, in the case of the cell lines, 1 day after seeding using Lipofectamine 2000 (Invitrogen, Carlsbad, CA) according to the manufacturer's protocol. Neurons in *Figures 3F, 4*, *Figure 3—figure supplement 1* and *Figure 4—figure supplement 1* were transfected using calcium phosphate as previously described (*Pavlos et al., 2010*). Immunostaining was then performed as described in *Chua et al. (2012)*. Briefly, cells were washed once with PBS to remove serum and then fixed using 4% paraformaldehyde. Afterwards, cells were permeabilized with 0.3% Triton-X-100 in PBS, rinsed with PBS and then blocked with 10% normal goat serum in PBS. Incubation with primary antibodies diluted in blocking solution was then carried out for 1 hr at room temperatures or overnight at 4°C. Subsequently, cells were exposed to secondary Cy3 or Alexafluor 488 conjugated goat anti-rabbit and anti-mouse antibodies, respectively, for 1 hr at room temperature. After washing, cells were mounted on slides (SuperFrost Plus, VWR International bvba, Leuven, Belgium) and then imaged using a confocal microscope (LSM 780, Zeiss, Germany) or an epifluorescence microscope (Axiovert 200M, Zeiss, Germany). Linescan analyses were performed using ImageJ or LAS AF Lite software.

## Labeling of recycled synaptic vesicles

To visualize synaptic vesicles that have undergone recycling, live neurons transfected with EGFP-Rab26WT were incubated in culture for 24 hr with Oyster 550-labeled anti-synaptotagmin-I antibodies (Synaptic Systems) that recognize its luminal domain (*Willig et al., 2006*). The neurons were then washed twice with PBS, fixed with 4% paraformaldehyde and analyzed under the microscope.

## Fly stocks and immunohistochemistry

The UAST-YFP.Rab26, UAST-YFP.Rab26Q250L, UAST-YFP.Rab26T204N (*Zhang et al., 2007*) and *elav*-Gal4 transgenic fly stocks were obtained from the Bloomington stock collection. Dissection and immunostaining of neuromuscular junctions from third instar larvae were performed as described (*Schmid and Sigrist, 2008*) using the following antibodies: mouse Anti-Brp (hybridoma clone nc82, DSHB; 1:50 dilution), anti-Csp antibody (hybridoma clone ab49, DSHB; 1:100 dilution), the chicken anti-GFP antibody (Abcam; 1:1000 dilution) and the goat anti-HRP (Sigma; 1:400 dilution). Dylight-649 labeled anti-goat and Alexa-488 labeled anti-chicken secondary antibodies were purchased from Jackson ImmunoResearch Laboratories (West Grove, PA). Alexa-568 conjugated anti-mouse secondary antibodies were purchased from Invitrogen (Carlsbad, CA). Images were acquired with a microscope (DMR-E; Leica, Germany) equipped with a scan head (TCS SP2 AOBS; Leica, Germany) and an oil-immersion 63 × 1.4 NA objective.

## Subcellular fractionation

Biochemical isolation of synaptic vesicles from the rat brain was performed as described previously (*Huttner et al., 1983*; *Takamori et al., 2006*). 12 µg of each brain fraction were loaded for analysis by immunoblotting to monitor the protein enrichment profile.

## Immunoisolation of synaptic vesicles

Purified monoclonal antibodies directed against Rab26 (described above) and synaptophysin (clone 7.2; [*Jahn et al., 1985*]) were coupled to epoxy-activated methacrylate microbeads (Eupergit C1Z, Röhm Pharmaceutical; note that these beads are no longer commercially available) and used for immunoisolation as described previously (*Burger et al., 1989*; *Takamori et al., 2000*). The bound vesicles were subsequently analyzed by electron microscopy or eluted with 40 µl 2 × SDS sample buffer for immunoblots analysis.

## Membrane extraction of Rab proteins with purified GDI

The RabGDI assay was performed as described in *Pavlos et al. (2010)*. Briefly, crude synaptic vesicles (LP2) were used as the starting material. 50 µg of LP2 were pre-incubated with 500 µM GDP or 500 µM GTPγS for 15 min at 37°C in 200 µl of extraction buffer containing 100 mM KCl, 5 mM MgCl$_2$, 10 mM EDTA, 50 mM HEPES-KOH pH 7.4 supplemented with a protease inhibitor cocktail (cOmplete EDTA-free, Roche, Mannheim, Germany). 5 µM of purified His-GDI were then added to each sample and further incubated for 45 min at 37°C. The samples were then kept on ice and subsequently centrifuged for 20 min at 200,000×*g*, 4°C using a Beckman S100 AT3 rotor. The resulting pellet was re-suspended in 50 µl 2 × lithium dodecyl sulfate (LDS) sample buffer (Invitrogen), boiled at 95°C for 5 min and analyzed by immunoblotting.

### Electron microscopy

For morphological analyses of immunoisolated vesicles, synaptophysin- or Rab26-conjugated microbeads containing the immunoisolated vesicles were first pelleted and subsequently immobilized using 2% agarose in 0.1 M cacodylate buffer at pH 7.4. Small agarose cubes containing the immobilized beads were fixed overnight at 4°C using 2% glutaraldehyde in 0.1 M cacodylate buffer at pH 7.4. After post-fixation in 1% osmium tetroxide and pre-embedding staining with 1% uranyl acetate, tissue samples were dehydrated and embedded in Agar 100. Thin sections (80 nm) were examined using a Philips CM 120 BioTwin transmission electron microscope (Philips Inc.Eindhoven, The Netherlands). Images were taken with a TemCam F224A slow scan CCD camera (TVIPS, Gauting, Germany). The evaluation of the samples was done using the iTEM software (Olympus Soft Imaging Solutions GmbH, Münster, Germany). For immunogold electron microscopy, ultrathin cryosections of neuronal cultures (*Figure 5A* and *Figure 5—figure supplement 1*) and HeLa cells (*Figure 6—figure*

*supplement 1*) transfected with EGFP-Rab26WT, were prepared as described previously (*Tokuyasu, 1973*, *1980*; *Zink et al., 2009*). For the ultrastructural analyses of the *Drosophila* neuromuscular junction (*Figure 5D*), a standard protocol was used. Briefly *Drosophila* filets were fixed by immersion using 2% glutaraldehyde in 0.1 M cacodylate buffer at pH 7.4 overnight at 4°C. After post-fixation in 1% osmium tetroxide and pre-embedding staining with 1% uranyl acetate, tissue samples were dehydrated and embedded in Agar 100. Thin sections (80 nm) were examined using a Philips CM 120 BioTwin transmission electron microscope (Philips Inc. Eindhoven, The Netherlands). Images were taken with a TemCam F416 slow scan CMOS camera (TVIPS, Gauting, Germany).

## Protein expression and purification

Human GST-tagged Rab26WT, Q123L, T77N were expressed in *Escherichia coli* BL21 (D3). 200 ml of pre-culture were grown at 37°C overnight. 10 ml of the pre-culture were then inoculated into 1 l of fresh LB medium supplemented with 100 µg/ml ampicillin and incubated for 2.5 hr at 37°C until the $OD_{600}$ reached a value of 0.6–0.8. The cultures were then incubated for 1 hr at 16°C. Induction was initiated by adding 1 mM IPTG to the cultures and the expression was carried out overnight at 16°C. Thereafter, cells were harvested by centrifugation at 4000 rpm for 10 min using a Beckman centrifuge. Pellets obtained from each 1 l culture flask were re-suspended in 25 ml of protein buffer containing 50 mM HEPES pH 7.4, 500 mM NaCl, 5 mM DTT, 5 mM $MgCl_2$, 100 µM GTPγS/GDP, supplemented with protease inhibitor cocktail and 1 mg/l of DNase. The samples were left for 10–15 min at 4°C and subsequently sonicated four times for 30 s each, separated by a 1 min incubation on ice, using a Branson Sonifier 450. The lysate was then cleared at 13,000 rpm using a SLA 1500 rotor for 40 min at 4°C. The resulting supernatant was collected and filtered using a 0.45 µm Whatman filter. The filtrate was then loaded onto a GST-column (GST Trap4B GE Healthcare, Germany) and eluted using 30 mM glutathione in protein buffer. The eluted fractions were collected and dialyzed three times for 3 hr each using protein buffer to remove glutathione. The purified proteins were then used for GST pulldown assays.

His-tagged murine Atg16L1(1–265)-pET-28a and His-tagged murine Atg5-pETDuet-1 were co-transformed into *E. coli* Rosetta 2 cells (Merck Millipore, Germany). Proteins were co-expressed in 3 l of ZYM-5052 autoinducing medium (*Studier, 2005*) supplemented with 100 mg/l ampicillin and 30 mg/l kanamycin for 3 hr at 37°C and followed by an overnight incubation at 18°C. Cells were harvested by centrifugation at 4500×*g* for 20 min. Pellets were resuspended in 100 ml buffer A (0.3 M NaCl, 1 mM $MgCl_2$, 35 mM imidazole, 50 mM $NaH_2PO_4$, pH 7.5). Cells were lysed by sonication and centrifuged for 1 hr at 25,000×*g*. The resulting supernatant was loaded onto two in line connected 1 ml HisTrap columns (GE Healthcare, Germany), washed with 80 ml buffer A and then eluted with a gradient from 0 to 100% over 80 ml of buffer B (0.3 M NaCl, 1 mM $MgCl_2$, 0.5 M imidazole, 50 mM $NaH_2PO_4$, pH 7.5). Fractions containing the purified proteins were pooled and dialyzed overnight at 4°C in gel filtration buffer (0.2 M NaCl, 1 mM $MgCl_2$, 25 mM HEPES, pH 7.5). The complex was concentrated and loaded onto a Superdex 200 16/60 column (GE Healthcare, Germany). Fractions were pooled and concentrated to 5 mg/ml, divided into aliquots, flash cooled in liquid nitrogen, and stored at −80°C.

## Co-immunoprecipitation and pulldown assays

Co-immunoprecipitation assays were performed as described in *Chua et al. (2012)*. Briefly, transiently transfected cells were washed once with ice-cold PBS and then lysed using lysis buffer (50 mM HEPES, 150 mM NaCl, 1 mM $MgCl_2$, 1% Tx-100 supplemented with a protease inhibitor cocktail) for 30 min. The lysate was then clarified by centrifugation at 10,000×*g* for 10 min. The resulting supernatant was incubated for 2 hr with anti-Flag or anti-GFP antibodies. Subsequently, 30 µl of protein G-Sepharose beads (GE Healthcare, Sweden) were added to each sample and further incubated for 1 hr under constant rotation. The samples were then washed thrice with lysis buffer. Finally, 25–30 µl of 2 × LDS sample buffer were then added to the beads and the mixture was boiled at 95°C for 5 min. 15 µl of the immunoprecipitated samples and 5 µl of the input were analyzed by immunoblotting.

For GST pulldown assays, 10 µg of each purified GST-Rab26 variant were first immobilized to 15 µl of Glutathione Sepharose 4 Fast Flow beads (GE Healthcare, Sweden) in a buffer containing 200 mM NaCl, 5 mM DTT, 5 mM $MgCl_2$, 10 µM GTPγS/GDP, 1% NP40, 50 mM HEPES, pH 7.4 for 30 min at 4°C with constant rotation. Subsequently, 10 µg of the pre-formed His-Atg16L1NT/His-Atg5FL complex was added to the mixture and further incubated for 1 more hr. The beads were then washed three times with buffer. 30 µl of 2 × LDS sample loading buffer was used to elute the proteins from the beads and 5 µl of each sample were subjected to SDS-PAGE and Western blotting.

## Analyses of Rab and effector interaction by analytical gel filtration

Human Rab26(54–233)Q123L was cloned into pET-28a using NdeI and XhoI cleavage sites. Murine Rab33B(30–202)Q92L (BC065076) was cloned into pETDuet-1 with BamHI and NotI restriction sites. Plasmids were transformed into *E. coli* BL21(DE3). Bacteria were grown in 3 l ZYM-5052 autoinducing medium supplemented with the appropriate antibiotic for 8 hr at 37°C. Cells were harvested by centrifugation and resuspended in 100 ml buffer A (30 mM imidazole, 0.2 M NaCl, 5 mM $MgCl_2$, 30 mM HEPES, pH 7.5). Bacteria were lysed with a microfluidizer and centrifuged for 1 hr at 25,000×*g*. Supernatant was loaded on a 1 ml HisTrap column (GE Healthcare, Germany) and washed with 25 ml buffer A and then eluted with a gradient from 0 to 100 % B over 20 ml (0.4 M imidazole, 0.2 M NaCl, 5 mM $MgCl_2$, 30 mM HEPES, pH 7.5). Fractions containing protein were pooled and diluted 1:1 with gel filtration buffer (0.15 M NaCl, 5 mM $MgCl_2$, 20 mM Hepes pH 7.5) and 250 µM GTP was added. Proteins were kept at 4°C overnight. Proteins were concentrated and loaded onto a Superdex 75 16/60 column (GE Healthcare, Germany). Fractions were then pooled and concentrated to at least 30 mg/ml, aliquoted and then flash cooled in liquid nitrogen and stored at −80°C.

For analytical gel filtration experiments 500 µl samples were loaded onto a Superdex 200 10/30 GL column (GE Healthcare, Germany) using an ÄktaPurifier (GE Healthcare, Germany). The gel filtration buffer was 0.15 M NaCl, 5 mM $MgCl_2$, 20 mM HEPES, pH 7.5. 80 µM Rab26(54–233)Q123L, 8 µM Atg16NT/Atg5 and 1 mM GTP were mixed, incubated for 20 min at RT and then loaded to the column. For comparison, 80 µM Rab33B(30–202)Q92L was similarly mixed with 8 µM Atg16NT/Atg5 and 1 mM GTP and analyzed. As controls, 8 µM Atg16NT/Atg5 alone, 80 µM Rab26(54–233)Q123L containing 1 mM GTP or 80 µM Rab33B(30–202)Q92L with 1 mM GTP were ran separately. Samples were analyzed on Coomassie stained 15% SDS PAGE gels.

## Acknowledgements

We thank Brigitte Barg-Kues and Sigrid Schmidt for excellent technical assistance.

## Additional information

### Competing interests

RJ: Reviewing editor, *eLife*. The other authors declare that no competing interests exist.

### Funding

| Funder | Grant reference number | Author |
| --- | --- | --- |
| National Health and Medical Research Council (NHMRC) | 463911 CJ Martin Fellowship | Nathan J Pavlos |

No formal funding was received for this work.

### Author contributions

BB, DR, GV, KK, Acquisition of data, Analysis and interpretation of data, Drafting or revising the article; NJP, Conception and design, Acquisition of data, Analysis and interpretation of data, Drafting or revising the article; DW, JB, Acquisition of data, Analysis and interpretation of data; AMS, CE, HM, Acquisition of data, Contributed unpublished essential data or reagents; JJC, RJ, Conception and design, Analysis and interpretation of data, Drafting or revising the article

### Author ORCIDs

Beyenech Binotti, http://orcid.org/0000-0001-7706-1096
Nathan J Pavlos, http://orcid.org/0000-0001-7003-188X
John JE Chua, http://orcid.org/0000-0002-5615-1014
Reinhard Jahn, http://orcid.org/0000-0003-1542-3498

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
