## [Decision Letter]

Thank you for sending your work entitled “Rab26 links synaptic vesicles to the autophagy pathway” for consideration at *eLife*. Your article has been favorably evaluated by Eve Marder (Senior editor) and 2 reviewers, one of whom is a member of our Board of Reviewing Editors.

The Reviewing editor and the other reviewer discussed their comments before we reached this decision, and the Reviewing editor has assembled the following comments to help you prepare a revised submission.

All reviewers were generally enthusiastic about your work. They felt that the data presented are of very high quality and generally support your conclusions, especially with regard to the SV association of Rab26. They also see strong potential in this work to represent a novel secretory vesicle degeneration pathway. However, the link to autophagy was considered still rather tenuous. While the massive somatic vesicle clustering and colocalization with Atg16L and LC3 may argue for a pre-autophagic event, the proof for a critical role of Rab26 for autophagy in neurons would require necessity/sufficiency experiments such as the impact of Rab26 loss of function block on induction of autophagy. We leave it up to you either to tone down your claim (Title, Discussion, etc) or to provide further experimental data.

Minor comments:

1) It is surprising why EGFP-Rab26 is not LC3 positive: is this a question of weaker efficiency or eGFP-Rab26? Please comment.

2) What role does Rab26 expression levels play in the morphological phenotype?

3) Figure 5: which synaptobrevin was used for colocalization with EGFP-Rab26?

---

## [Author Response]

*All reviewers were generally enthusiastic about your work. They felt that the data presented are of very high quality and generally support your conclusions, especially with regard to the SV association of Rab26. They also see strong potential in this work to represent a novel secretory vesicle degeneration pathway. However, the link to autophagy was considered still rather tenuous. While the massive somatic vesicle clustering and colocalization with Atg16L and LC3 may argue for a pre-autophagic event, the proof for a critical role of Rab26 for autophagy in neurons would require necessity/sufficiency experiments such as the impact of Rab26 loss of function block on induction of autophagy. We leave it up to you either to tone down your claim (Title, Discussion, etc) or to provide further experimental data*.

We agree that we do not have conclusive evidence showing directly that Rab26 activation triggers autophagy of synaptic vesicles. The problem, as far as we see it, is that in situ this pathway is probably not active or of only very low activity, making loss-of-function experiments rather difficult. We have tried various approaches but it became clear that considerable additional efforts (including suitable animal models) are required for obtaining air-tight evidence, and it will probably be more efficient to carry out these experiments with a laboratory having the required tools and experiences.

We have therefore revised the Discussion as suggested in order to make sure we are not over-interpreting our results or draw conclusions that are not substantiated by the evidence. We have also revised the Abstract accordingly. As far as the Title is concerned, we could not come up with something better: “link”, at least by our understanding, means a connection (which we have shown) and does not imply necessity/sufficiency. Therefore, we have retained it except for a small change suggested by the *eLife* office, but we are open for a better suggestion.

Minor comments:

*It is surprising why EGFP-Rab26 is not LC3 positive: is this a question of weaker efficiency or eGFP-Rab26? Please comment*.

We were also puzzled by the observation that EGFP-Rab26, unlike FLAG-Rab26, does not readily localize with LC3. Possibly, the addition of the bulky EGFP-moiety (which is of similar size as the GTPase itself) interferes with the binding of additional factors needed for recruiting LC3 or for targeting vesicles to LC3 positive compartments, thereby trapping the vesicles at a step just prior to the recruitment of LC3. While uncommon, there are notable examples of the bulky GFP moiety interfering with protein-protein interactions and thus with function such as in the ESCRT pathway (Howard et al., 2001, J. Cell Sci., 114, 2395; Langelier et al., 2006, J. Virol. 80, 9465; Strack et al., 2003, Cell, 114, 689; Tandon et al., 2009, J. Virol., 83, 10797). Prevention of LC3-recruitment may therefore render the Rab26-induced vesicle clusters “stuck” as they cannot progress to form mature autophagosomes that are then degraded. Such a scenario would explain the accumulation of the enormous vesicle clusters observed within neuronal cell bodies that whilst retained Atg16L1, formed no or only incomplete isolation membranes. It may also account for the densely packed vesicle clusters lacking isolation membranes observed upon expression of EGFP-Rab26 in *Drosophila* neuromuscular synapses.

*What role does Rab26 expression levels play in the morphological phenotype*?

A higher level of Rab26 WT expression in cells (monitored by immunocytochemistry) does increase the phenotype. Nevertheless, even though Rab26 WT and Rab26 QL both express at similar levels in these cells (see e.g. Figure 2), the WT variant still exhibits the stronger morphological phenotype. Consequently, the functional state of Rab26 (rather than its level alone) is the dominant factor influencing the magnitude of the phenotype.

Figure 5*: which synaptobrevin was used for colocalization with EGFP-Rab26*?

Antibodies specific for synaptobrevin 2 were used. This information has been added to the legend.